# Design and Synthesis of New Modified Flexible Purine Bases as Potential Inhibitors of Human PNP

**DOI:** 10.3390/molecules28030928

**Published:** 2023-01-17

**Authors:** Anastasia Khandazhinskaya, Ilja Fateev, Barbara Eletskaya, Anna Maslova, Irina Konstantinova, Katherine Seley-Radtke, Sergey Kochetkov, Elena Matyugina

**Affiliations:** 1Engelhardt Institute of Molecular Biology, Russian Academy of Sciences, 119991 Moscow, Russia; 2Shemyakin-Ovchinnikov Institute of Bioorganic Chemistry, Russian Academy of Sciences, 117997 Moscow, Russia; 3Department of Chemistry & Biochemistry, University of Maryland, Baltimore County, Baltimore, MD 21250, USA

**Keywords:** aza/deazapurines, fleximer inhibitor, human purine nucleoside phosphorylase

## Abstract

The great interest in studying the structure of human purine nucleoside phosphorylase (*h*PNP) and the continued search for effective inhibitors is due to the importance of the enzyme as a target in the therapy of T-cell proliferative diseases. In addition, *h*PNP inhibitors are used in organ transplant surgeries to provide immunodeficiency during and after the procedure. Previously, we showed that members of the well-known fleximer class of nucleosides are substrates of *E. coli* PNP. Fleximers have great promise as they have exhibited significant biological activity against a number of viruses of pandemic concern. Herein, we describe the synthesis and inhibition studies of a series of new fleximer compounds against *h*PNP and discuss their possible binding mode with the enzyme. At a concentration of 2 mM for the flex-7-deazapurines **1–4**, a decrease in enzymatic activity by more than 50% was observed. 4-Amino-5-(1H-pyrrol-3-yl)pyridine **2** was the best inhibitor, with a Ki = 0.70 mM. Docking experiments have shown that ligand **2** is localized in the selected binding pocket Glu201, Asn243 and Phe200. The ability of the pyridine and pyrrole fragments to undergo rotation around the C–C bond allows for multiple binding modes in the active site of *h*PNP, which could provide several plausible bioactive conformations.

## 1. Introduction

Human purine nucleoside phosphorylase (*h*PNP) belongs to a family of enzymes involved in the purine salvage pathway of nucleoside biosynthesis, which promotes the utilization of purine bases [1]. Its primary role is to catalyze the cleavage of inosine, 2’-deoxyinosine, guanosine and 2’-deoxyguanosine (dG) to the corresponding base and sugar-1-phosphate via reversible phosphorolysis. PNP deficiency in humans leads to a decrease in the number of T cells and immunodeficiencies. Therefore, *h*PNP is considered an attractive target in the therapy of T-cell proliferative diseases, primarily T-cell leukemias and lymphomas, as well as psoriasis, multiple sclerosis, rheumatoid arthritis, etc. [2,3]. In addition, *h*PNP inhibitors are used in organ transplant surgeries to provide an immunodeficiency status during and after the procedure. In some cases, *h*PNP inhibitors are also able to enhance the activity of antiviral and antitumor nucleoside drugs [1,4] by inhibiting the premature metabolism of the nucleoside drug. All of these reasons explain the great interest in studying the structure of human purine nucleoside phosphorylase and the continued search for more effective inhibitors.

Most inhibitors of purine nucleoside phosphorylases are structural analogues of nucleoside substrates, modified in the nucleobase and/or the carbohydrate moiety. Using rational design, Schramm et. al. [5] discovered analogues of C-nucleosides—known as immucillins—which proved to be inhibitors for the transition state of PNP [6]. Several immucillins have successfully completed or are currently in various stages of clinical trials, including Immucillin-H (Forodesine^®^, Mundesine^®^), which was approved in Japan for treating relapsed or refractory peripheral T-cell lymphoma [7], and DADMe-immucillin H (DADMe-Immucillin-H, BCX4208, Ulodesine), which passed the second stage of clinical trials in the treatment of gout [8]. High levels of inhibitory activity were demonstrated by acyclic analogues of immucillins, namely DATMe-Immucillin-H and SerMe-Immucillin-H [1]. It should also be noted that many of the known *h*PNP inhibitors are derivatives of deazapurine bases.

We previously showed that fleximer analogues of aza/deazapurine heterocyclic bases are substrates of *E. coli* purine nucleoside phosphorylase [9] while their 5′-norcarbocyclic derivatives are noncompetitive inhibitors of the same enzyme [10]. Both groups of compounds are members of the well-known fleximer class of nucleosides. Fleximers are modified nucleosides where the bicyclic heterocyclic base is split into two separate moieties connected by a C–C bond [11]. This modification introduces additional flexibility into the molecule, which facilitates new structural interactions with the target enzyme. Fleximers have shown great promise and have exhibited significant biological activity against a number of viruses of pandemic concern [12,13,14]. 

Analysis of the literature on PNP inhibitors showed that aza and deaza analogues of purine bases, as well as nucleoside analogues with increased flexibility, including fleximeric and acyclic ones, have high potential in terms of PNP inhibitory properties. Thus, we designed and synthesized two new groups of compounds, in particular, fleximer analogues of 7-deazapurines and acyclic derivatives of previously synthesized pyrazole-containing bases. Herein, we describe the synthesis and inhibition studies of the new compounds against *h*PNP and discuss their possible binding modes with the enzyme.

## 2. Results and Discussion

### 2.1. Chemistry

To obtain new inhibitors of *h*PNP, we designed and synthesized two groups of modified fleximer bases. The first comprised pyrrole-containing fleximer bases **1–4,** which were chosen due to the inhibitory activity exhibited by 7-deazahypoxanthine against *E. coli* PNP (Ki = 0.13 mM) [15]. Similarly, the acyclic 8-aza-7-deazapurine fleximers **13–16** were designed based on the structure of previously studied 5′-norcarbocyclic 8-aza-7-deazapurine fleximers, which showed weak inhibitory activity against *E. coli* PNP (Ki = 14–20 mM) [10] and acyclovir, which was able to inhibit *h*PNP (Ki = 90µM) [16]. 

The fleximer analogues of 7-deazapurine bases **1–4**, containing pyrrole, as a five-membered heterocyclic moiety, and 2-aminopyridine, 4-aminopyridine/pyrimidine or 4(3H)-pyrimidone, as a six-membered heterocyclic component, were synthesized (Figure 1) using Suzyki–Miyaura cross-coupling from commercially available N-TIPS pyrrole-3-boronic acid pinacol ester **5** and corresponding bromides **6–9 [17]**. The triisopropylsilyl group was removed with tetrabutylammonium fluoride in THF and the pivaloyl groups by reflux in methanol with K_2_CO_3_. Target analogues **1–4** were obtained in moderate yields (Figure 1).

The second group of compounds synthesized was the acyclic fleximer analogues of 8-aza-7-deazapurine bases. As shown in Figure 2, acyclic fleximers **13–16** were obtained using Vorbruggen conditions, as described previously [13]. Instead of the classical use of ribose tetraacetate, 2-acetoxyethylacetoxymethyl ester **17** was introduced (Figure 2). 

For the synthesis of acyclic analogues **22–25**, protected fleximer bases **18–21 [9]** were refluxed in 1,1,1,3,3,3-hexamethyldisilazane (HMDS) with ammonium sulfate and pyridine for 4 h (Figure 2). After evaporation of the solvents, the residue was dissolved in acetonitrile followed by addition of trimethylsilyl trifluoromethanesulfonate (TMSOTf) and 2-acetoxyethylacetoxymethyl ester **17**. The acetyl and pivaloyl protecting groups of compounds **22–24** were removed simultaneously by reflux in MeOH with K_2_CO_3_. Products **13–15** were obtained in a 69–74% yield. The acetyl group of compound **25** was removed with 7N ammonia in methanol. Further hydrogenation with 10% palladium−charcoal catalyst gave analogue **16** with a 14% yield.

### 2.2. Inhibition Studies

Compounds **1–4** and **13–16** were then studied as potential inhibitors of *h*PNP (Figure 1). At a concentration of 2 mM for flexible analogues of 7-deazapurines **1–4**, a decrease in enzymatic activity by more than 50% was observed. Flex-acyclovir analogues **13–16** did not appear to be inhibitors, although at a concentration of 0.5 mM, acyclovir slightly reduced the rate of inosine phosphorolysis, as reported by Rabuffetti et al. [1]. 

The inhibition constants of compounds **1–4** are shown in Table 1, which reveal that the most pronounced inhibition occurred in the presence of compound **2**. The flex-hypoxanthine analogue **1** showed better binding to the enzyme active site than the flex-adenine analogue **4**. The inhibition constant for 7-deazaguanine (7DG) is 0.2 mM, which is lower than the constants for the investigated compounds [18].

Non-competitive inhibition was observed in all cases (Appendix A, see Appendix A). This may be caused by the formation of a dead-end complex upon the binding of the heterocyclic base and an inorganic phosphate at the active site of the human enzyme. This was seen for various heterocyclic bases, including 7-deazahypoxanthine, in the case of *E. coli* PNP [10].

### 2.3. Molecular Modeling

Compound **2** was chosen for computational studies. Before docking, four crystallographic structures of human purine nucleoside phosphorylases from the Protein Database were analyzed. This was a necessary step to determine the most frequent atomic interactions (hydrogen bonds, π–π stacking, etc.) of the PNP active site with various ligands [19]. The following complexes were chosen: with the natural substrate inosine [20] and with inhibitors, including Immucillin H, 7-deazaguanine (7DG) [18] and acyclovir [16] (Appendix A, see Appendix A). Using the ProteinsPlus web service (https://proteins.plus accessed on 15 December 2022) and a special PoseView tool, two-dimensional diagrams of the selected complexes were created [21,22]. 

Note that, in all cases, the same patterns were responsible for the binding of the purine heterocyclic base and its orientation at the site of enzyme binding. The following amino acid residues of the protein side chains were identified: hydrogen bonding with Glu201 and Asn243, and π–π stacking with Phe200. It is, therefore, assumed that the inhibition of *h*PNP by flex-base **2** occurs due to interaction with these amino acid residues (Figure 2). 

Structures of flex-base **2** were generated and structural optimizations were performed by using HyperChem software [23]. Quantum chemical analysis of **2** shows that the pyrrole and pyridine fragments are located in different planes (are non-coplanar).

Docking of **2** was then performed in a limited area in a volume of 12 Å × 10 Å × 10 Å, which is sufficient to accommodate the ligand inside the receptor. After clustering, the resulting ligand–protein complexes were ranked according to the established binding energies. The states with the lowest energy and specific conservative ligand–protein interactions were selected as potentially possible. Interactions of protein–ligands were measured using the BIOVIA Discovery Studio Visualizer 2021.

Docking experiments showed that **2** is localized in the selected binding pocket. The free energy of binding (ΔG) is listed in Table 2. 

Three models were chosen from various positions of the fleximer base 2 in the hPNP active site. In the selected models, a similar interaction pattern is observed in the binding site: the base is positioned in the same plane as 7DG, one of the aromatic rings is oriented perpendicular to the Phe200 side residue and the amino group (at the pyridine ring) and the proton at nitrogen (pyrrole ring) participate in the formation of hydrogen bonds with Glu201 and Asn243.

Model A is characterized by the minimum energy ΔG. Model B is similar to the arrangement of 7DG in crystal structure of the *h*PNP [18]. However, model B is less energetically favorable than models A and C. It is somewhat difficult to assess the exact position of **2** in the active site, which leads to the inhibitory activity. In our opinion, due to the flexibility of flex-base **2**, various binding options are possible in the *h*PNP active site and inhibition of the enzyme is presumably realized in one of the three variants presented.

## 3. Materials and Methods

Reactions were performed using commercially available reagents (Acros, Aldrich and Fluka). Anhydrous solvents were prepared according to standard procedures. Column chromatography was carried out on silica gel Kieselgel 60 (40–60 µm, straight phase) (3 × 20 cm) (Merck, Germany). Thin-layer chromatography (TLC) was performed on TLC Silica gel 60 F25 plates. The human PNP recombinant protein (Q01) was provided by Biozol Diagnostica Vertrieb GmbH (Eching, Germany). Human PNP stock solution: 0.15 mg/mL.

NMR spectra were recorded on Bruker Avance III spectrometer (Bruker BioSpin, Rheinstetten, Germany) with an operating frequency of 300 MHz for ^1^H-NMR and 75.5 MHz for ^13^C-NMR in CDCl_3_, CD_3_OD or DMSO-d6.

High-resolution mass spectra (HRMS) were obtained on a Bruker Daltonics micrOTOF-Q II instrument using electrospray ionization (ESI). The measurements were acquired in a negative ion mode with the following parameters: interface capillary voltage—3700 V; mass range from m/z 50 to 3000; external calibration (Electrospray Calibrant Solution, Fluka); nebulizer pressure—0.3 Bar; flow rate—3 µL/min; dry gas nitrogen (4.0 L/min); interface temperature was set at 180 or 190 °C. A syringe injection was used.

### 3.1. Chemistry

#### 3.1.1. General Method for the Synthesis of Compounds **10–12** and 5-(1H-pyrrol-3-yl)pyrimidin-4(3H)-one **1**

To the solution of compounds **6–9** (0.5–2.7 mmol) in DME, Pd(PPh_3_)_4_ (5mol%) was added. The mixture was stirred for 15 min. Then, N-TIPS pyrrole-3-boronic acid pinacol ester (1.5 eq) dissolved in DME and saturated NaHCO_3_ was added. The reaction mixture was refluxed for 4 h at room temperature. The progress of the reaction was monitored using TLC. The solvents were evaporated and the residue was dissolved in CHCl_3_ and extracted with water. Organic layer was washed with brine, dried using Na_2_SO_4_ and solvent was evaporated. Crude products were dissolved in THF and TBAF (1.1 eq) was added. After 5 min, the solvent was evaporated and the residues were purified using column chromatography on silica gel with an appropriate elution system.

4-Pivaloylamino-5-(1H-pyrrol-3-yl)pyridine (**10**). Compound **10** was synthesized from 3-bromo-4-pivaloylaminopyridine **7** (500 mg, 2.6 mmol) and N-TIPS pyrrole-3-boronic acid pinacol ester **5** (1 g, 2.8 mmol). After purification on a silica gel column eluting with chloroform: methanol (95:5), **10** was obtained as a yellow powder. Yield 210 mg, 63%. ^1^H NMR (300 MHz, Methanol-d4) δ 8.38 (1H, s), 8.32 (2H, s), 7.02–7.01 (2H, m), 6.33 (1H, dd, J = 2.5, 1.9 Hz), 1.22 (9H, s). ^13^C NMR (75 MHz, Methanol-d4) δ 177.9, 149.0, 147.2, 143.2, 123.3, 119.8, 117.2, 115.1, 113.7, 107.4, 39.8, 26.2. 

2-Pivaloylamino-3-(1H-pyrrol-3-yl)pyridine (**11**). Compound **11** was synthesized from 3-bromo-2-pivaloylaminopyridine **8** (225 mg, 0.9 mmol) and N-TIPS pyrrole-3-boronic acid pinacol ester **5** (456 mg, 1.3 mmol). After purification on a silica gel column using chloroform: methanol (95:5) system for elution, the yield of compound **11** as pale-yellow powder was 71% (150 mg). ^1^H NMR (300 MHz, Chloroform-d) δ 9.74 (1H, s), 8.41 (1H, dd, J = 4.9, 1.9 Hz,), 7.70 (1H, dd, J = 7.6, 1.9 Hz), 7.12 (1H, dd, J = 7.6, 5.0 Hz), 6.99–6.95 (2H, m), 6.35–6.33 (1H, m), 1.25 (9H, s). ^13^C NMR (75 MHz, Chloroform-d) δ 176.3, 148.5, 145.3, 138.6, 124.2, 120.0, 119.8, 118.3, 117.3, 107.9, 40.1, 27.5.

4-Pivaloylamino-5-(1H-pyrrol-3-yl)pyrimidine (**12**). Compound **12** was synthesized starting from 4-pivaloylamino-5-bromopyrimidine **9** (520 mg, 2.68 mmol) and N-TIPS pyrrole-3-boronic acid pinacol ester **5** (1 g, 2.95 mmol). The reaction was purified on a silica gel column using chloroform: methanol (95:5) system for elution and **12** was obtained as white powder with 31% (200 mg) yield. ^1^H NMR (300 MHz, Chloroform-d) δ 9.48 (1H, s), 8.95 (1H, s), 8.56 (1H, s), 8.48 (1H, s), 7.10–6.96 (2H, m), 6.39–6.36 (1H, m), 1.26 (9H, s). ^13^C NMR (75 MHz, Chloroform-d) δ 175.7, 156.5, 156.3, 155.4, 120.5, 117.5, 114.7, 107.9, 27.3.

5-(1H-pyrrol-3-yl)pyrimidin-4(3H)-one (**1**). Compound **1** was synthesized starting from 5-bromopyrimidin-4(3H)-one **6** (100 mg, 0.57 mmol) and N-TIPS pyrrole-3-boronic acid pinacol ester **5** (300 mg, 0.86 mmol). The reaction was purified on a silica gel column using chloroform: methanol (9:1) system for elution and **1** was obtained as white powder in a 15% (24mg) yield. ^1^H NMR (300 MHz, DMSO-d6) δ 12.45 (1H, s), 10.91 (1H, s), 8.23 (1H, s), 7.97 (1H, s), 7.64–7.62 (1H, m), 6.80–6.78 (1H, m), 6.58–6.56 (1H, m). ^13^C NMR (75 MHz, DMSO-d6) δ 160.0, 146.7, 145.9, 123.5, 119.5, 118.8, 115.5, 105.5. HRMS m/z: calculated for C_8_H_7_N_3_O [M+H]^+^ 162.0662; found [M+H]^+^ 162.0662. 

#### 3.1.2. General Method for the Synthesis of Compounds **2–4**

To a solution of fleximer base analogues **10–12** (0.6–0.8 mmol) in methanol, K_2_CO_3_ (2 eq) was added. The reaction mixture was refluxed for 3–4 h. The progress of the reaction was monitored using TLC. Solvent was evaporated and the residues were purified using column chromatography on silica gel with appropriate elution system. 

4-Amino-5-(1H-pyrrol-3-yl)pyridine (**2**). Compound **2** was synthesized starting from compound **10** (210 mg; 0.87 mmol). Purification on a silica gel column using chloroform: methanol (8:2) system for elution gave **2** (110 mg) as an off-white powder. Yield 27%. ^1^H NMR (300 MHz, Methanol-d4) δ 8.04 (1H, s), 7.90 (1H, d, J = 5.9 Hz), 6.99 (1H, t, J = 1.8 Hz), 6.91 (1H, dd, J = 2.8, 1.9 Hz), 6.74–6.72 (1H, m), 6.33 (1H, dd, J = 2.7, 1.6 Hz). ^13^C NMR (75 MHz, Methanol-d4) δ 153.5, 145.3, 143.8, 118.8, 118.3, 116.3, 116.2, 108.8, 106.9. HRMS m/z: calculated for C_9_H_9_N_3_ [M+H]^+^ 160.0869; found [M+H]^+^ 160.0868.

2-Amino-3-(1H-pyrrol-3-yl)pyridine (**3**). Compound **3** was synthesized from compound **11** (150 mg; 0.62 mmol). Purification on a silica gel column using chloroform: methanol (8:2) system for elution gave **3** (76 mg) as a pale-yellow powder. Yield 48% ^1^H NMR (300 MHz, DMSO-d6) δ 11.02 (1H, s), 7.81 (1H, dd, J = 4.9, 1.8 Hz), 7.42 (1H, dd, J = 7.3, 1.9 Hz), 7.07 (1H, q, J = 2.1 Hz), 6.87 (1H, q, J = 2.4 Hz), 6.59–6.57 (1H, m), 6.33 (1H, q, J = 2.5 Hz), 5.47 (2H, s). ^13^C NMR (75 MHz, DMSO-d6) δ 156.7, 145.1, 135.8, 119.9, 119.1, 116.8, 116.6, 113.7, 107.4. HRMS m/z: calculated for C_9_H_9_N_3_ [M+H]^+^ 160.0869; found [M+H]^+^ 160.0867.

4-Amino-5-(1H-pyrrol-3-yl)pyrimidine (**4**). Compound **4** was synthesized starting from compound **12** (200 mg; 0.82 mmol). Purification on a silica gel column using chloroform: methanol (9:1) system for elution gave **4** (100 mg) as a white powder. Yield 23%. ^1^H NMR (300 MHz, Methanol-d4) δ 8.30 (1H, s), 8.08 (1H, s), 6.96 (1H, t, J = 1.8 Hz), 6.89 (1H, dd, J = 2.8, 1.9 Hz), 6.33 (1H, dd, J = 2.8, 1.7 Hz). ^13^C NMR (75 MHz, Methanol-d4) δ 154.8, 152.0, 119.6, 116.6, 115.2, 107.0. HRMS m/z: calculated for C_8_H_8_N_4_ [M+H]^+^ 161.0822; found [M+H]^+^ 161.0822. 

Compounds **18–21** were synthesized as described earlier [9]. 

#### 3.1.3. General Method for the Synthesis of Compounds **22–25**

Compounds **18–21** (1 mmol) were refluxed in 1,1,1,3,3,3-hexamethyldisilazane (20 mL) with 50 mg of ammonium sulfate and pyridine (2 mL) for 4 h. Then, solvents were evaporated to dryness and dried under high vacuum. The residues were dissolved in acetonitrile (15 mL) followed by addition of trimethylsilyl trifluoromethanesulfonate (TMSOTf) (1.5 mmol) and (2-acetoxyetoxy)methyl acetate (1 mmol). The reaction mixture was stirred for 18 h. The progress of the reaction was monitored by TLC in the system. The product was purified by column chromatography on silica gel eluting with chloroform: methanol (98:2) or hexane: ethyl acetate (1:4) system. 

4-Pivaloylamino-5-(1-((2-acetoxyethoxy)methyl)-pyrazol-4-yl)pyridine (**22**). Compound **22** was synthesized starting from compound **18** (61 mg; 0.25 mmol), TMSOTf (68 µL; 0.38 mmol) and (2-acetoxyetoxy)methyl acetate (40 µL; 0.25 mmol). Purification on a silica gel column using chloroform:methanol (99:1, then 97:3) system gave **22** (59 mg) as colorless oil. Yield 66%. ^1^H NMR (300 MHz, Methanol-d4) δ 8.57–8.45 (1H, m), 8.48–8.36 (1H, m), 8.16 (1H, d, J = 0.8 Hz), 8.02 (1H, d, J = 5.5 Hz), 7.85 (1H, d, J = 0.7 Hz), 5.60 (2H, s), 4.23–4.13 (2H, m), 3.86–3.71 (2H, m), 3.37 (1H, s), 2.03 (3H, s), 1.26 (9H, s). ^13^C NMR (75 MHz, Methanol-d4) δ 178.1, 171.2, 149.6, 148.2, 143.4, 139.4, 130.1, 117.3, 116.0, 80.4, 67.0, 62.9, 39.6, 29.3, 26.2, 19.3.

2-Pivaloylamino-5-(1-((2-acetoxyethoxy)methyl)-pyrazol-3-yl)pyridine (**23**). Compound **23** was synthesized starting from compound **19** (61 mg; 0.25 mmol), TMSOTf (68 µL; 0.38 mmol) and (2-acetoxyetoxy)methyl acetate (40 µL; 0.25 mmol). Purification on a silica gel column using chloroform:methanol (99:1, then 97:3) system gave **23** (62 mg) as colorless oil. Yield 69%. ^1^H NMR (300 MHz, Methanol-d4) δ 8.40 (1H, dd, J = 4.8, 1.8 Hz), 8.05 (1H, d, J = 0.8 Hz), 7.99 (1H, dd, J = 7.8, 1.8 Hz), 7.81 (1H, d, J = 0.8 Hz), 7.43 (1H, dd, J = 7.7, 4.9 Hz), 5.54 (2H, s), 4.21–4.13 (2H, m), 3.78–3.68 (2H, m), 2.03 (3H, s), 1.28 (9H, s). ^13^C NMR (75 MHz, Methanol-d4) δ 179.1, 171.2, 146.7, 139.1, 138.7, 129.6, 126.8, 123.0, 119.2, 80.2, 66.9, 62.8, 38.7, 29.3, 26.3, 19.3.

4-Pivaloylamino-5-(1-((2-acetoxyethoxy)methyl)-pyrazol-4-yl)pyrimidine (**24**). Compound **24** was synthesized starting from compound **20** (49 mg; 0.2 mmol), TMSOTf (55 µL; 0.3 mmol) and (2-acetoxyetoxy)methyl acetate (32 µL; 0.2 mmol). Purification on a silica gel column using chloroform:methanol (99:1, then 97:3) system gave **24** (57 mg) as colorless oil. Yield 79%. ^1^H NMR (300 MHz, Methanol-d4) δ 8.91 (1H, s), 8.82 (1H, s), 8.15 (1H, d, J = 0.8 Hz), 7.84 (1H, d, J = 0.8 Hz), 5.56 (2H, s), 4.82 (2H, s), 4.22–4.10 (2H, m), 3.80–3.71 (2H, m), 2.03 (3H, s), 1.27 (9H, s). ^13^C NMR (75 MHz, Methanol-d4) δ 177.9, 171.2, 157.2, 156.0, 138.8, 129.7, 80.3, 66.9, 62.8, 39.3, 26.24, 26.0, 19.3.

4-Benzyloxy-5-(1-((2-acetoxyethoxy)methyl)-1H-pyrazol-4-yl)pyrimidine (**25**). Compound **25** was synthesized starting from compound **21** (252 mg; 1 mmol), TMSOTf (333 µL; 1.5 mmol) and (2-acetoxyetoxy)methyl acetate (176 µL; 1 mmol). Purification on a silica gel column using hexane: ethyl acetate (1:4) system gave **25** (150 mg) as colorless oil. Yield 40%. ^1^H NMR (300 MHz, Chloroform-d) δ 8.49 (1H, s), 8.29 (1H, s), 8.25 (1H, s), 7.98 (1H, s), 5.51 (2H, s), 5.22 (2H, s), 4.19–4.16 (2H, m), 3.75–3.71 (2H, m), 2.06 (3H, s). ^13^C NMR (75 MHz, Chloroform-d) δ 170.9, 159.0, 148.3, 145.8, 137.7, 134.9, 130.3, 129.2, 128.6, 128.2, 120.3, 115.0, 80.8, 67.1, 63.0, 50.4, 20.8.

4-Benzyloxy-5-(1-((2-hydroxyethoxy)methyl)-1H-pyrazol-4-yl)pyrimidine (**26**). Compound **25** (100 mg; 0.27 mmol) was dissolved in NH_3_/MeOH and left at 37 ^0^C for 3 h. Then, solvent was evaporated and the residue used in the following reaction without purification. ^1^H NMR (300 MHz, Methanol-d4) δ 8.51 (1H, s), 8.46 (1H, s), 8.36 (1H, s), 7.42–7.29 (5H, m), 5.54 (2H, s), 5.26 (2H, s), 3.65–3.56 (4H, m). ^13^C NMR (75 MHz, Methanol-d4) δ 159.4, 149.5, 147.0, 138.0, 135.9, 130.3, 128.5 (2C), 127.9, 127.7 (2C), 80.4, 70.4, 60.5, 49.8.

#### 3.1.4. General Method for the Synthesis of Compounds **13–15**

For solution of acyclic fleximer analogues **22–24** (0.15 mmol) in methanol (3 mL), K_2_CO_3_ (0.4 mmol) was added. The reaction mixture was refluxed for 3–4 h. The progress of the reaction was monitored using TLC. Solvent was evaporated and the residues were purified using column chromatography on silica gel with appropriate elution system. 

4-Amino-5-(1-((2-hydroxyethoxy)methyl)-pyrazol-4-yl)pyridine (**13**). Compound **13** was synthesized starting from compound **22** (55 mg; 0.15 mmol). Purification on a silica gel column using chloroform: methanol (8:2) system for elution gave **13** (25 mg) as an off-white powder. Yield 71%. ^1^H NMR (300 MHz, Methanol-d4) δ 8.09 (2H, t, J = 1.8 Hz), 7.97 (1H, d, J = 5.9 Hz), 7.79 (1H, s), 6.76 (1H, d, J = 5.9 Hz), 5.58 (2H, s), 3.65–3.62 (4H, m). ^13^C NMR (75 MHz, Methanol-d4) δ 153.4, 146.8, 145.8, 139.0, 129.5, 116.4, 109.3, 80.5, 70.6, 60.5, 29.32. HRMS m/z: calculated for C_11_H_14_N_4_O_2_ [M+H]^+^ 235.1190; found [M+H]^+^ 235.1183.

2-Amino-3-(1-((2-hydroxyethoxy)methyl)-1H-pyrazol-4-yl)pyridine (**14**). Compound **14** was synthesized starting from compound **23** (55 mg; 0.15 mmol). Purification on a silica gel column using chloroform: methanol (8:2) system for elution gave **14** (24 mg) as an off-white powder. Yield 69%. ^1^H NMR (300 MHz, Methanol-d4) δ 8.11 (1H, s), 7.90 (1H, dd, J = 5.2, 1.8 Hz), 7.83 (1H, s), 7.57 (1H, dd, J = 7.4, 1.8 Hz), 6.74 (1H, dd, J = 7.4, 5.1 Hz), 5.57 (2H, s), 3.73–3.56 (5H, m). ^13^C NMR (75 MHz, Methanol-d4) δ 156.4, 145.3, 138.9, 137.6, 129.2, 119.0, 113.7, 80.5, 70.5, 60.5, 29.3. HRMS m/z: calculated for C_11_H_14_N_4_O_2_ [M+H]^+^ 235.1190; found [M+H]^+^ 235.1190. 

4-Amino-5-(1-((2-hydroxyethoxy)methyl)-1H-pyrazol-4-yl)pyrimidine (**15**). Compound **15** was synthesized starting from compound **24** (54 mg; 0.15 mmol). Purification on a silica gel column using chloroform:methanol (8:2) system for elution gave **15** (26 mg) as an off-white powder. Yield 74%. ^1^H NMR (300 MHz, Methanol-d4) δ 8.35 (1H, s), 8.15–8.13 (2H, m), 7.83 (1H, s), 5.58 (2H, s), 3.69–3.62 (4H, m). ^13^C NMR (75 MHz, Methanol-d4) δ 161.5, 156.0, 152.5, 138.8, 129.6, 115.1, 110.4, 80.5, 70.6, 60.5, 29.3. HRMS m/z: calculated for C_10_H_13_N_5_O_2_ [M+H]^+^ 236.1142; found [M+H]^+^ 236.1137.

5-(1-((2-hydroxyethoxy)methyl)-1H-pyrazol-4-yl)pyrimidin-4(3H)-one (**16**). Compound **26** (0.27 mmol) was dissolved in methanol and 10 % Pd/C was added. Then, reaction mixture was treated with H_2_ for 2 h. The progress of the reaction was monitored using TLC. The reaction mixture was filtered through celite and solvent was evaporated. The residue was purified using preparative chromatography glass plate in chloroform:methanol (8:2) system. The yield of **16** was 14% (9 mg). ^1^H NMR (300 MHz, Methanol-d4) δ 8.51 (1H, s), 8.38 (1H, s), 8.14–8.12 (2H, m), 5.55 (2H, s), 3.66–3.57 (4H, m). ^13^C NMR (75 MHz, Methanol-d4) δ 163.5, 147.2, 137.9, 130.0, 114.9, 80.4, 70.5, 60.5, 35.5, 29.3. HRMS m/z: calculated for C_10_H_12_N_4_O_3_ [M+H]^+^ 237.0982; found [M+H]^+^ 237.0980. 

### 3.2. Enzyme Inhibition

Each reaction mixture (100 μL, 50 mM KH_2_PO_4_, pH 7.0) contained human purine nucleoside phosphorylase (0.003 μg), 0.2 mM inosine and 2 mM of tested compounds **1–4**, **13–16**. For kinetic parameters: each reaction mixture (50 μL, 50 mM KH_2_PO_4_, pH 7.0) contained human purine nucleoside phosphorylase (0.003 μg), inosine (0.005–2 mM) and inhibitor (0, 0.5, 1 or 2 mM of **1**, **2**, **3** or **4**). Specific activity of the PNP is 160 μmol/min*mg. Reaction mixtures were incubated for 2 min at 37 °C. Substrate and product quantities were determined using HPLC (Waters 1525, column Ascentis Express C18, 2.7 μm, 3.0 × 75 mm, eluent A 0.1% aqueous TFA, eluent B 0.1% TFA/70% acetonitrile in water, detection at 254 nm, UV-detector Waters 2489). Each experiment was repeated three times. Kinetic parameters were determined by nonlinear regression analysis using SciDAVis v2.3.0 software. Simple equation for non-competitive inhibition was used. V = Vmax*S/[(KM+S)*(1+Ci/Ki)].

### 3.3. Molecular Modelling and Docking Studies

Computational docking studies were conducted on protein–ligand docking server SwissDock, based on EADock DSS [24,25]. Structures of fleximer molecules were generated and structural optimizations were performed by using HyperChem software [23]. The ab initio amber FF 6–31G** method; RMS values 0.3 kcal/mol. Analysis of molecular structures and conformational searching from related docking data were performed using the UCSF Chimera program [26].

A crystallographic model of the human purine nucleoside phosphorylase was published and posted on the Protein Data Bank website (PDB ID code 3INY). It is the cryo-EM structure of the *h*PNP complex bound to its inhibitor 7-deazaguanine [18,27]. 

To confirm the docking result, a ‘pose selection’ method was selected to re-dock 7-deazaguanine with the active site PNP [28]. Values of the root-mean-square deviation of atomic positions (RMSD), docking pose, accuracy and coverage of contacts were compared with the co-crystallized structure. RMSD of atomic coordinates between two molecules was calculated using the PyMOL Molecular Graphics System, Version 2.5, Schrodinger, LLC. The obtained value has acceptable range of RMSD docking < 1.5 Å [13]. RMSD = 0.112 Å (279 to 279 atoms) (see Appendix A).

## 4. Conclusions

We synthesized two new groups of compounds, namely fleximer analogues of 7-deazapurines and acyclic derivatives of pyrazole-containing bases. The compounds were studied as potential inhibitors of *h*PNP. At a concentration of 2 mM for the flex-7-deazapurines **1–4**, a decrease in enzymatic activity by more than 50% was observed. 4-Amino-5-(1H-pyrrol-3-yl)pyridine (**2**) was the best inhibitor, with a Ki = 0.70 mM. Flex-acyclic analogues **13–16** did not appear to be inhibitors. In order to understand the possible binding mode of the most active compound **2** and the enzyme, computer modeling was performed, which could also aid in additional investigations of *h*PNP inhibitors. Structures of the flex-base **2** were generated and structural optimizations were performed by using HyperChem soft. Quantum chemical analysis of **2** showed that the pyrrole and pyridine fragments are located in different planes (are non-coplanar). Docking experiments revealed that **2** is localized in the selected binding pocket interacting with Glu201, Asn243 and Phe200. The ability of the pyridine and pyrrole fragments to undergo rotation around the C–C bond allows for multiple binding options in the active site of *h*PNP, which could provide several plausible bioactive conformations. 

## Data Availability

The data presented in this study are available in the article and Appendix A.

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
