# Peer review of "Design and Synthesis of New Modified Flexible Purine Bases as Potential Inhibitors of Human PNP"

_molecules, 2023, doi:10.3390/molecules28030928_

Round 1

Reviewer 1 Report

The authors describe the  synthesis of some purine bases as potential inhibitors of Human PNP. Overall, manuscript is fine, however, some minor corrections are required:

-Structure of compound 5 should  be improved in scheme 1.

-In scheme 1, conditions c), temperature should be mentioned. Same required for d) in scheme 2.

-In line 46, 'Shram' is spelled differently to the name provided in ref 5.

-'7-deazapurine bases' is spelled differently in lines 88 and 91.

-English language needs to be improved. Proper use of punctuation is lacking in the manuscript. Authors need to carefully check it.

-In Materials and Methods, synthesis of compounds should be mentioned as per journal's format using sub-headings and making the sub-heading bold.

Author Response

The authors appreciate the reviewer’s comments and added all the required data.

The authors describe the  synthesis of some purine bases as potential inhibitors of Human PNP. Overall, manuscript is fine, however, some minor corrections are required:

-Structure of compound 5 should be improved in scheme 1.

The structure was improved.

-In scheme 1, conditions c), temperature should be mentioned. Same required for d) in scheme 2.

The temperature was added.

-In line 46, 'Shram' is spelled differently to the name provided in ref 5.

Corrected.

-'7-deazapurine bases' is spelled differently in lines 88 and 91.

Corrected.

-English language needs to be improved. Proper use of punctuation is lacking in the manuscript. Authors need to carefully check it.

Checked and improved.

-In Materials and Methods, synthesis of compounds should be mentioned as per journal’s format using sub-headings and making the sub-heading bold.

Improved according to the journal format.

Reviewer 2 Report

In this paper, Matyugina and co-workers report the synthesis of two new groups of molecules and describe their inhibition studies against hPNP, a potential target in the therapy of T-cell proliferative diseases. Here some corrections and suggestions to take into consideration.

- Correct structures 13-16 and 22-26, as they are, it seems that a methylene group from 2-acetoxyethylacetoxymethyl ester 17 is missing.

- Please include chemical yields directly in schemes 1 and 2.

- Since it is stated in the title, a well-described rational design section of the new enzyme inhibitors should be included.

- High resolution mass spectra (HRMS) analysis is missing for some compounds.

Author Response

The authors appreciate the reviewer’s comments and have made the corrections.

In this paper, Matyugina and co-workers report the synthesis of two new groups of molecules and describe their inhibition studies against hPNP, a potential target in the therapy of T-cell proliferative diseases. Here some corrections and suggestions to take into consideration.

- Correct structures 13-16 and 22-26, as they are, it seems that a methylene group from 2-acetoxyethylacetoxymethyl ester 17 is missing.

The structure of compound 17 was corrected.

- Please include chemical yields directly in schemes 1 and 2.

Chemical yields were included.

- Since it is stated in the title, a well-described rational design section of the new enzyme inhibitors should be included.

Reason for the design of new compounds – inhibitors of PNP has been included to the introduction.

- High resolution mass spectra (HRMS) analysis is missing for some compounds.

HRMS was added to the supplementary.